# SNIP: Bridging Mathematical Symbolic and Numeric Realms with Unified Pre-training

**Kazem Meidani**[* 1]**, Parshin Shojaee**[* 2]**,**
**Chandan K. Reddy** [2]**, Amir Barati Farimani** [1,3]
[1] Department of Mechanical Engineering, Carnegie Mellon University
[2] Department of Computer Science, Virginia Tech
[3] Machine Learning Department, Carnegie Mellon University

## Abstract

In scientific inquiry, symbolic mathematical equations play a fundamental role in modeling complex natural phenomena. Leveraging the power of deep learning, we introduce SNIP, a Multi-Modal Symbolic-Numeric Pre-training framework. By employing joint contrastive learning between symbolic and numeric domains, SNIP enhances their mutual alignment in pre-trained embeddings. Latent space analysis reveals that symbolic supervision significantly enriches the embeddings of numeric data, and vice versa. Evaluations across diverse tasks, including symbolic-to-numeric and numeric-to-symbolic property prediction, demonstrate SNIP's superior performance over fully supervised baselines. This advantage is particularly pronounced in few-shot learning scenarios, making SNIP a valuable asset in situations with limited available data.

## 1   Introduction

Throughout the history of science, symbolic mathematics has been unreasonably effective in representing natural phenomena [1]. Complex patterns of natural systems, represented as numeric data observations, can be elegantly abstracted using mathematical formulas. Mathematical symbolism has given us the language to describe, understand, and predict the natural world. The challenge of bridging the gap between the numeric observations and their mathematical symbolic representations has been a consistent focus in many scientific and engineering domains. Recognizing and exploring this connection is crucial, as it promises to drive advancements in various fields.

In recent years, deep learning has demonstrated promising capabilities in learning from symbolic mathematics language as well as extracting knowledge from numeric data observations. Transformer-based models [2], in particular, have emerged as frontrunners in this endeavor, effectively capturing patterns within mathematical expressions and solving complex tasks such as differential equations [3, 4]. Efforts have also been made to enhance the mathematical reasoning of language models, improving their performance in general math word problem solving [5, 6]. However, these models, while powerful, are not inherently designed to handle numeric data observations. While some pre-trained symbolic regression models have been introduced to map numeric datasets to their governing mathematical expressions in a supervised manner [7, 8], a gap still remains in developing a task-agnostic unified pre-training model capable of mutual understanding between the modalities of symbolic mathematical equations and their corresponding numeric counterparts.

Multi-modal pre-training models, exemplified by groundbreaking models like Contrastive Language-Image Pre-training (CLIP) [9], have found a significant place in the deep learning landscape. CLIP has particularly set new standards in vision-language tasks, bridging the understanding

---

*Equal contribution. Contact email: mmeidani@andrew.cmu.edu

NeurIPS 2023 AI for Science Workshop.

between visual content and natural language descriptions. This mutual comprehension across different data modalities has opened up opportunities for more intuitive and context-aware machine learning applications. Expanding beyond traditional vision-language domains, recent studies have broadened multi-modal pre-training to include other modalities, such as audio and tabular data [10, 11, 12]. Additionally, previously untouched scientific domains, like molecular representation, are also benefiting from these advancements [13, 14]. Nevertheless, the symbolic-numeric domain remains relatively unexplored. Considering the foundational role of symbolic mathematics in science and the ubiquity of numeric data, an in-depth exploration of their mutual learning is not only timely but essential. Such an investigation holds the promise of unlocking numerous applications, from improving scientific simulations and modeling to enhancing data-driven decision-making in diverse sectors.

In this work, we present **S**ymbolic-**N**umeric **I**ntegrated **P**re-training (**SNIP**) to connect the two often distinct worlds of symbolic mathematical expressions and their corresponding numeric manifestations. The architecture of SNIP, depicted in Fig. 1, incorporates dual transformer-based encoders, with each encoder dedicated to learning the symbolic or numeric representations of mathematical functions. Subsequently, a task-agnostic joint contrastive objective is employed to enhance the similarity between (symbolic, numeric) pairs of data. The unified multi-modal pre-training of SNIP provides capabilities to understand and generate cross-modal content. Our experiments show that SNIP achieves remarkable performance in cross-modal mathematical property understanding and prediction tasks. Analysis of the latent embeddings further unveils that SNIP's pre-trained representations manifest discernible patterns associated with these cross-modal properties. Moreover, when paired with an equation generation decoder and upon deeper exploration of the latent space, we observe that SNIP's representations are interpolatable. This suggests a significant relationship between the latent vectors and their numeric behaviors. To put it simply, latent space interpolation translates to a symbolic function whose numeric behavior semantically bridges the gap between the original and target functions (check Fig.4 for more details). Such latent space dynamics, ingrained through SNIP's multi-modal pre-training, offer a valuable edge for sophisticated searches and explorations, potentially benefiting various subsequent downstream tasks.

## 2   Related Work

**Large-scale Pre-training.** Our work is built upon an extensive body of research advocating the advantages of pre-training large models on large datasets [15, 16]. Initially, pre-training was single-modal, with self-supervised learning (SSL) as a key paradigm that used data as its own supervision, especially useful where labeled data was limited [17]. This paved the way for the emergence of multi-modal pre-training, where models are trained to understand relationships across different data types [18]. Vision and language have traditionally played the two main characters of pre-training models. For instance, CLIP [9], ALIGN [19], and FLAVA [20] utilize image-caption pairs to construct jointly learned embedding spaces. These models are trained to align the embeddings of corresponding image-caption pairs while distancing unrelated pairs. The success of multi-modal pre-training in vision and language spurred its adoption in other domains. For example, recent works have extended this approach to videos, audio, and even tabular data [10, 21, 12]. Specialized scientific domains have also embraced this paradigm. For instance, different models have emerged to learn joint representations of molecules [13, 14]. Our work introduces a fresh perspective, intertwining symbolic mathematics with numeric observations. To this end, we use multi-modal pre-training's potential to deepen the symbolic-numeric mutual understanding.

**Deep Symbolic Mathematics.** Recently, deep learning models have made significant strides in the field of mathematical reasoning [6, 5]. The Transformer architecture, originally designed for NLP tasks [2], has been repurposed with remarkable success in the realm of symbolic mathematics. It has powered models that can integrate functions [3, 4], prove mathematical theorems [22], and perform numerical calculations, such as arithmetic operations [23, 24]. These achievements underscore the flexibility and potential of deep learning models in abstract reasoning. Beyond pure symbolic reasoning, there is also a growing interest in supplementing these models with numerical knowledge for improved mathematical understanding. For example, recent endeavors have sought to enhance language models with numeric representations, aiming to improve their skills in mathematical word problem-solving [25, 26, 27, 28]. Our work contributes a new angle to this growing field by integrating symbolic and numeric understanding in a unified pre-training framework. By doing so, we not

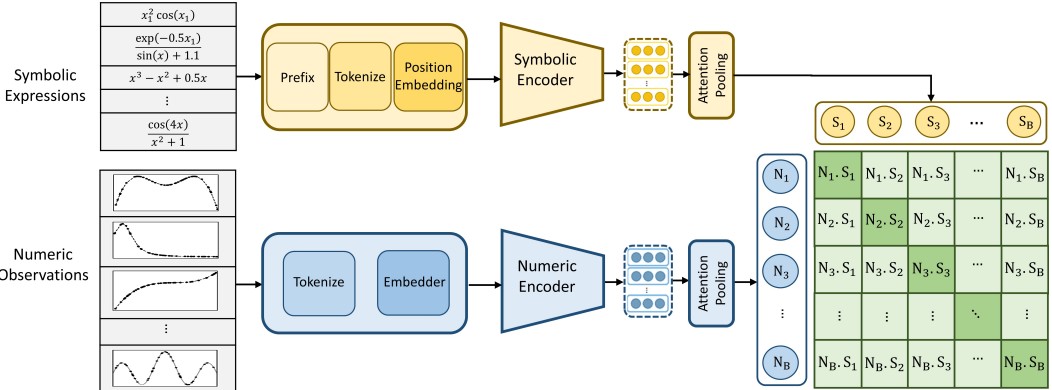

Figure 1: The SNIP Framework: A schematic representation of the dual-encoder pre-training scheme for mutual learning between symbolic expressions and their numeric data observations. Both symbolic and numeric encoders work in tandem, capturing the paired similarities and essence of their respective modalities.

only capture the abstract representations of mathematical symbolic concepts but also their tangible numeric behaviors.

## 3 Pre-training

As depicted in Fig. 1, the SNIP architecture comprises two transformer-based encoders, each tailored for learning the symbolic or numeric representations of mathematical functions. These symbolic and numeric encoders are jointly trained with a task-agnostic joint contrastive objective to predict correct pairings within a batch of (symbolic, numeric) examples. During pre-training, SNIP receives synthetically created symbolic equations and their associated numeric data as inputs to the symbolic and numeric heads, respectively.

### 3.1 Numeric Encoder

The numeric encoder's foundation is rooted in the recent advancements of transformer-based models for encoding numeric observations into latent spaces [8, 29, 7]. In this framework, the numeric encoder—represented as $\mathcal{E}_\theta^V$—integrates an embedder, a multi-layer Transformer, and an attention pooling approach, to map numeric observations $(\boldsymbol{x}, \boldsymbol{y})$ into a condensed latent vector $\boldsymbol{Z}_V$.

**Tokenization.** Following [8, 29], numeric inputs are tokenized using base-10 floating-point notation. They are rounded to four significant digits and subsequently represented as sequences of three tokens: sign, mantissa (0-9999 range), and exponent ($E$-100 to $E$100). For instance, the number $5.432$ is tokenized as $[+, 5432, E\text{-}3]$.

**Encoding.** Given a batch of $N$ numeric input points $(\boldsymbol{x}, \boldsymbol{y}) \in \mathbb{R}^{D+1}$, each is represented by $3(D + 1)$ tokens. With increasing $D$ and $N$, the input sequence length grows, challenging the quadratic complexity of Transformers. To address this, we employ an embedder, as suggested by [8], before the Transformer encoder. This embedder maps each input point to a unique embedding space. The resulting embeddings, with dimension $d_{\text{emb}}$, are then fed into the encoder. For the numeric encoder, we utilize a multi-layer Transformer architecture [2]. Notably, due to the permutation invariance of the $N$ input points for each batch sample, we exclude positional embeddings, aligning with the approach in [8]. This encoder variant is denoted as $Enc^V$. The representation at its $l$-th layer is given by $\boldsymbol{V}_l = Enc_l^V(\boldsymbol{V}_{l-1})$, where $l$ ranges from 1 to $L_V$, and $L_V$ signifies the total layer count.

**Attention-based Distillation.** To distill the information from the Transformer's output into a compact representation for the whole sequence of observations, we employ an attention-based pooling mechanism, following [30]. Let $\mathcal{A}_V$ denote the attention weights, which are computed as: $\mathcal{A}_V = \text{softmax}\left(\boldsymbol{W}_a \cdot \boldsymbol{V}_{L_V}^T\right)$, where $\boldsymbol{W}_a \in \mathbb{R}^{d_{\text{emb}}}$ is a learnable weight matrix, and we take the transpose of $\boldsymbol{V}_{L_V} \in \mathbb{R}^{N \times d_{\text{emb}}}$ to apply softmax along the sequence dimension $N$. The compact sequence-level representation, $\boldsymbol{Z}_V$, is then obtained by: $\boldsymbol{Z}_V = \mathcal{A}_V \cdot \boldsymbol{V}_{L_V}$. This attention mechanism allows the model to focus on the most informative parts of the data points, effectively compressing the information into a fixed-size embedding.

## 3.2 Symbolic Encoder

The symbolic encoder in our framework also draws inspiration from recent advancements in transformer-based models for encoding symbolic mathematical functions, as demonstrated in works such as [4, 3]. Here, the symbolic encoder—denoted as $\mathcal{E}_\psi^S$—is a composite entity parameterized by $\psi$, encapsulating the embedder, a multi-layer Transformer, and attention-based pooling mechanisms. Given an input symbolic function $f(\cdot)$, this encoder outputs a condensed representation $\boldsymbol{Z}_S$.

**Tokenization.** Mathematical functions are tokenized by enumerating their trees in prefix order, following the principles outlined in [8]. This process employs self-contained tokens to represent operators, variables, and integers, while constants are encoded using the same methodology as discussed in Sec. 3.1, representing each with three tokens. In alignment with [3], we use special tokens [$\langle BOS \rangle$] and [$\langle EOS \rangle$] to mark sequence start and end.

**Encoding.** Given a batch of symbolic functions with $M$ tokens, each symbolic input is represented as $\boldsymbol{S}_0 = \big[ \boldsymbol{E}_{[\langle BOS \rangle]}; \boldsymbol{E}_{t_1}; \ldots; \boldsymbol{E}_{t_M}; \boldsymbol{E}_{[\langle EOS \rangle]} \big] + \boldsymbol{S}^{pos}$, where $\boldsymbol{S}_0 \in \mathbb{R}^{(M+2) \times d_{\text{emb}}}$. Here, $\boldsymbol{E}$ refers to the embedding matrix, $t_i$ denotes the $i$-th token, $M$ signifies the number of tokens in the symbolic function, $d_{\text{emb}}$ is the embedding dimension, and $\boldsymbol{S}^{pos}$ represents the positional embedding matrix. In the symbolic encoder, we use a Transformers model with the same architecture settings as in Sec. 3.1. This variant of the encoder, denoted as $Enc^S$, processes the input symbolic data. The $l$-th layer representation is described as $\boldsymbol{S}_l = Enc_l^S(\boldsymbol{S}_{l-1})$, where $l$ varies from 1 to $L_S$, and $L_S$ indicates the total number of layers within the symbolic encoder.

**Attention-based Distillation.** The symbolic encoder also employs attention-based pooling, as in Sec. 3.1. This mechanism computes weighted sums to distill information from the symbolic expression into a compact representation $\boldsymbol{Z}_S = \mathcal{A}_S \cdot \boldsymbol{S}_{L_S}$, using attention weights $\mathcal{A}_S$ through softmax along the symbolic sequence.

## 3.3 Unified Pre-training Objective

Our work introduces a unified symbolic-numeric pre-training approach, SNIP, which aims to facilitate a mutual understanding of both domains, enabling advanced cross-modal reasoning.

**Training Objective.** SNIP's pre-training objective is inspired by the joint training used in CLIP [9]. Incorporating both a numeric and symbolic encoder, the model optimizes a symmetric cross-entropy loss over similarity scores. It employs a contrastive loss (InfoNCE [31] objective) to learn the correspondence between numeric and symbolic data pairs. Specifically, this approach learns to align embeddings of corresponding symbolic-numeric pairs while distancing unrelated pairs. The objective function can be defined as:

$$\mathcal{L} = - \sum_{(v,s) \in B} \big( \log \text{NCE}(\boldsymbol{Z}_S, \boldsymbol{Z}_V) + \log \text{NCE}(\boldsymbol{Z}_V, \boldsymbol{Z}_S) \big), \tag{1}$$

where $B$ represents the batch of (symbolic, numeric) data pairs, $\text{NCE}(\boldsymbol{Z}_S, \boldsymbol{Z}_V)$ and $\text{NCE}(\boldsymbol{Z}_V, \boldsymbol{Z}_S)$ denote the contrastive losses on symbolic-to-numeric and numeric-to-symbolic similarities, respectively. The symbolic-to-numeric contrastive loss, $\text{NCE}(\boldsymbol{Z}_S, \boldsymbol{Z}_V)$, is calculated as follows:

$$\text{NCE}(\boldsymbol{Z}_S, \boldsymbol{Z}_V) = \frac{\exp \big( \boldsymbol{Z}_S \cdot \boldsymbol{Z}_V^+ \big)}{\sum_{\boldsymbol{Z} \in \{\boldsymbol{Z}_V^+, \boldsymbol{Z}_V^-\}} \exp \big( \frac{\boldsymbol{Z}_S \cdot \boldsymbol{Z}}{\tau} \big)} \tag{2}$$

Here, $\tau$ is temperature, $\boldsymbol{Z}_V^+$ represents positive SNIP numeric embeddings that overlap with SNIP symbolic embedding $\boldsymbol{Z}_S$, and $\boldsymbol{Z}_V^-$ are negative numeric embeddings implicitly formed by other numeric embeddings in the batch. A symmetric equivalent, $\text{NCE}(\boldsymbol{Z}_V, \boldsymbol{Z}_S)$, also defines the numeric-to-symbolic contrastive loss. More implementation details are provided in App. B.

## 3.4 Pre-training Data

In our SNIP approach, pre-training relies on a vast synthetic dataset comprising paired numeric and symbolic data. We follow the data generation mechanism in [8], where each example consists of $N$ data points $(x, y) \in \mathbb{R}^{D+1}$ and a corresponding mathematical function $f$, where $y = f(x)$. Data generation proceeds in several steps, ensuring diverse and informative training examples. More details about each of the following steps are provided in App. A.

**Sampling of functions.** We create random mathematical functions using a process detailed in [8, 3]. This process involves selecting an input dimension $D$, determining the number of binary operators,

constructing binary trees, assigning variables to leaf nodes, inserting unary operators, and applying random affine transformations. This method ensures a diverse set of functions for training.

**Sampling of datapoints.** After generating a function, we sample $N$ input points and find their corresponding target values. To maintain data quality, we follow guidelines from [8], discarding samples with inputs outside the function's domain or exceptionally large output values. Our approach includes drawing inputs for each function from various distributions, enhancing training diversity. The generation process of datapoints also involves selecting cluster weights and parameters, sampling input points for each cluster, and normalization along each dimension. To emphasize on the function's numeric behavior rather than the range of values, we also normalize the target values $y$ between $(0, 1)$.

## 4    Pre-trained Latent Space Analysis

**Symbolic Encoded Representations.** To evaluate the learned representations of SNIP, we analyze the pre-trained latent space to investigate the mutual understanding that is achieved between the symbolic and numeric representations. We first show that numeric behaviors are learned in the symbolic latent vectors $\boldsymbol{Z}_S$. To this end, we introduce several mathematical properties that describe different numeric features of the mathematical functions. Specifically, we consider the following properties: (a) *Non-Convexity Ratio (NCR)* which approximates function convexity with values between NCR=0 (fully convex) and NCR=1 (fully concave); (b) *Upwardness* which quantifies the function's directionality by assessing the segments where data increases within the training domain, ranging from UP=-1 for strictly decreasing functions to UP=1 for increasing ones; (c) *Average of Normalized y* can be a measure to distinguish

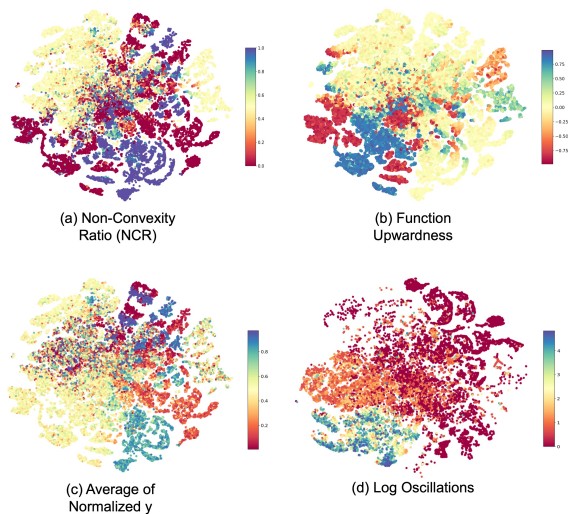

(a) Non-Convexity Ratio (NCR)

(b) Function Upwardness

(c) Average of Normalized y

(d) Log Oscillations

Figure 2: 2D t-SNE visualizations of the latent space of Symbolic encoded vectors $\boldsymbol{Z}_S$ of pre-trained SNIP, colored by **(a)** Non-Convexity Ratio, **(b)** Upwardness, **(c)** Average of normalized $y$, and **(d)** Oscillations (Logarithmic scale).

different numeric behaviors, and it can roughly approximate the numerical integral of the normalized function in the defined range of training $\boldsymbol{x}$; and (d) *Log Oscillations* which quantifies the degree of oscillatory behavior exhibited by the numeric data, represented in logarithmic scale. More details of these properties can be found in App. C.

Fig. 2 illustrate two-dimensional t-SNE [32] visualizations of SNIP symbolic latent space colored by these properties. We can observe that the latent spaces are shaped by the symbolic-numeric similarities of the functions such that numeric properties can be clustered and/or show visible trends in the symbolic encoded representation space $\boldsymbol{Z}_S$.

**Numeric Encoded Representations.** Just as numeric behaviors shaped symbolic encoded representations, numeric vectors, denoted as $\boldsymbol{Z}_V$, are similarly influenced by the symbolic characteristics inherent to the associated governing equations. This relationship is visually depicted in Fig. 3, which presents 2D t-SNE visualizations of the latent space cultivated from SNIP's numeric vectors. These visualizations are color-coded to reflect two specific symbolic attributes: (a) complexity of the function, and (b) a predetermined classification based on

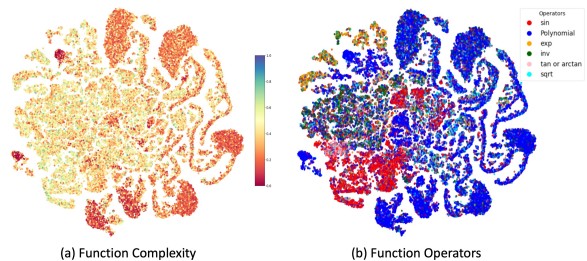

(a) Function Complexity

(b) Function Operators

Figure 3: 2D t-SNE plots of the latent space of Numeric encoded vectors $\boldsymbol{Z}_V$ of pretrained SNIP on 1D datasets, colored by **(a)** Function Complexity, and **(b)** Function Classes based on Operators.

dominant operators in the functions. The *Function Complexity* refers to the function's length when represented in prefix order notation, effectively counting the number of nodes in its expression tree. On the other hand, *Function Operator Categorization* is a broader classification that groups functions based on the predominant operators present in their symbolic mathematical expressions. These operators not only affect the function's behavior but also provide insights into the nature of the data they represent. It's crucial to recognize that a function may encompass several operators, adding layers to the intricacy of the data's behavior. Moreover, specific operators within a function might have a pronounced impact, dictating the data's scope and pattern. More details of these symbolic attributes and categorization are provided in App. C.

**Latent Space Interpolation.** To delve deeper into the interpolation capabilities of SNIP's pretrained latent representations, we paired the SNIP encoder with an equation generation decoder. This fusion enabled us to explore the latent space in greater depth, unveiling the interpolatability inherent in SNIP's representations. The notion of *interpolatability*, as vividly illustrated in Fig.4, speaks to a profound association between the latent space embeddings and their corresponding numeric behaviors. In the presented figure, we start with a source function, represented by the numeric encoded vector $\boldsymbol{Z}_V^s$ (visualized as the blue curve). We then select a destination function, represented by $\boldsymbol{Z}_V^d$ (depicted as the orange curve). A linear interpolation is carried out between these numeric encoded vectors to derive an intermediate representation, $\boldsymbol{Z}_V^{int}$. When this interpolated latent vector is decoded, we obtain a symbolic function represented as $\hat{f}$. Evaluating $\hat{f}$ over the dataset $\boldsymbol{x}$ reveals a fascinating insight: *the interpolated function manifests behavior that semantically bridges the gap between the behaviors of the source and destination functions*. Such latent space dynamics, ingrained through SNIP's multimodal pre-training, offer a valuable edge for sophisticated searches and explorations, potentially benefiting various subsequent downstream tasks.

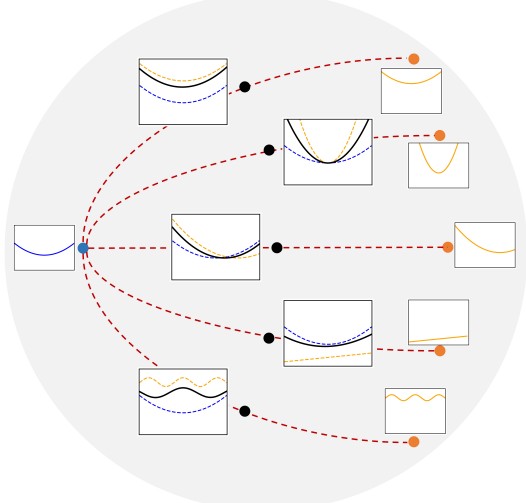

Figure 4: Interpolatability of SNIP numeric latent space.

## 5    Using SNIP for Cross-modal Property Prediction

To further evaluate SNIP's capability for cross-modal comprehension between symbolic and numeric domains, we conducted targeted experiments. These tests aimed to assess the model's aptitude for predicting specific mathematical properties from one domain based on insights from the other—a non-trivial task requiring mutual understanding of both. Due to space limitations, only results for *NCR* and *Upwardness*, as described in section 4, are discussed here. More experiments and SNIP's pre-trained representations are provided in App. D.

### 5.1    Models and Training

To assess property prediction using SNIP's embeddings, we employ a predictor head that passes these embeddings through a single-hidden-layer MLP to yield the predicted values. We adopt a Mean Squared Error (MSE) loss function for training on continuous properties. We consider three key training configurations to probe the efficacy of SNIP's learned representations:

• **Supervised Model**: Utilizes the same encoder architecture as SNIP but initializes randomly.

• **SNIP (frozen)**: Keeps the encoder parameters fixed, training only the predictor head.

• **SNIP (finetuned)**: Initializes encoder from pretrained SNIP, allowing full updates during training.

For a fair comparison, all model variants are trained on identical datasets comprising 10K equations and subsequently tested on a distinct 1K-equation evaluation dataset. These datasets are generated using the technique described in Sec. 3.4, as per [8].

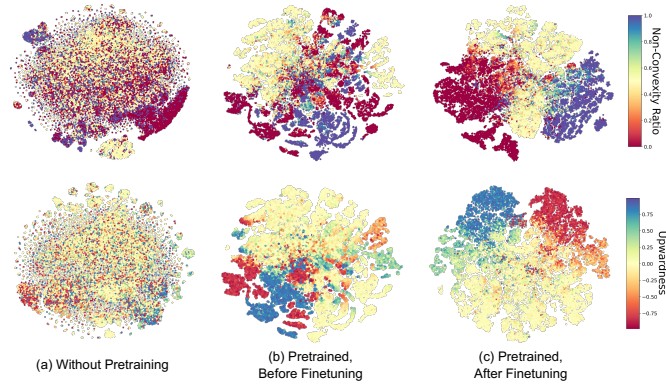

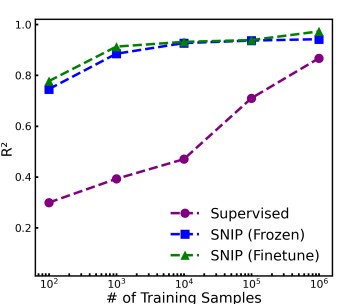

| | | |
|---|---|---|
| (a) Without Pretraining | (b) Pretrained, Before Finetuning | (c) Pretrained, After Finetuning |

Figure 5: 2D t-SNE representations of the encoded vectors across three model variants, colored for **(top)** Non-Convexity Ratio and **(bottom)** Function Upwardness prediction tasks.

Figure 6: $R^2$ scores for *NCR* property prediction task vs. the number of training samples.

## 5.2 Results

**Quantitative Results.** Table 1 presents the $R^2$ and Normalized Mean Squared Error (NMSE) for all three models across the tasks of predicting *NCR* and *Upwardness*. Results reveal a significant gap in performance between the purely supervised model and those benefiting from SNIP's prior knowledge. This performance gap can be attributed to SNIP's pre-trained, semantically rich representations, enabling enhanced generalization to unseen functions. Additionally, fine-tuning the SNIP encoder results in marginal performance gains, indicating the model's capability to adapt to specific downstream tasks.

Table 1: Results of using SNIP for property prediction.

| Model | Non-Convexity Ratio | | Upwardness | |
|---|---|---|---|---|
| | $\uparrow R^2$ | $\downarrow$ NMSE | $\uparrow R^2$ | $\downarrow$ NMSE |
| Supervised | 0.4701 | 0.5299 | 0.4644 | 0.5356 |
| SNIP (frozen) | 0.9269 | 0.0731 | 0.9460 | 0.0540 |
| SNIP (finetuned) | **0.9317** | **0.0683** | **0.9600** | **0.0400** |

**Qualitative Findings.** To delve deeper into the power of SNIP's representations, we compared its pre-finetuning and post-finetuning latent spaces against that of a supervised model lacking pretraining, using t-distributed Stochastic Neighbor Embedding (t-SNE) [32]. The visualizations are color-coded by the corresponding properties (Fig. 5). Consistent with the quantitative outcomes, the supervised model's latent space, shown in Fig. 5(a), exhibits limited structural coherence. In contrast, SNIP's latent space in Fig. 5(b) shows pronounced clustering and distinct property trends. Notably, further fine-tuning of the encoder for these prediction tasks, depicted in Fig. 5(c), results in a more structured latent space, marked by clearer linear trends in properties. This finding underscores SNIP's quantitative advantages and its flexibility in adapting to downstream tasks.

**Few-shot Learning Analysis.** We evaluated how training sample size influences the test $R^2$ scores for predicting *NCR*, assessing three model variants on a fixed 1K-sample test set (Fig. 6). In few-shot scenarios with just 100 training samples, the supervised model's score fell sharply to 0.292, while both SNIP variants maintained scores above 0.745. At 10K training samples, SNIP's performance advantage remained consistent. Upon increasing the training sample size to 1M, all models showed improvement; the supervised model notably increased its score to 0.867. Yet, both fine-tuned and frozen SNIP variants continued to lead, posting scores of 0.973 and 0.942, respectively. These results emphasize SNIP's superior generalization from limited data, underscoring the SNIP's rich semantic encodings.

## 6 Discussion and Conclusion

We introduced SNIP, a multi-modal symbolic-numeric pre-training model that learns how to associate the symbolic and numeric aspects of mathematical functions. We showed that SNIP exhibits remarkable few-shot capabilities in estimating cross-modal mathematical properties, outperforming fully-supervised models. While SNIP showcases robustness and versatility in integrating symbolic and numeric learning, it has notable limitations. It struggles with data patterns that cannot be clearly expressed as closed-form mathematical functions. Also, its performance is tied to the

pre-defined data generation protocol, adopted from [3, 8], which sets constraints on factors such as the vocabulary of mathematical operators. Despite these limitations, SNIP has a wide range of capabilities, presenting a powerful tool in the intersection of symbolic and numeric mathematics. Looking ahead, SNIP offers a rich foundation for numerous advancements. Future research can harness numeric guidance to enhance symbolic-to-symbolic tasks like function integration. Conversely, symbolic insights might enhance numeric-to-numeric tasks, such as zero-shot extrapolation and super-resolution. The mutual symbolic and numeric understandings within SNIP could also open doors for complex multi-modal tasks, notably in numeric-to-symbolic equation generation or symbolic regression. Furthermore, the embeddings learned by SNIP present an opportunity to craft novel metrics for evaluating symbolic-numeric proximity and to develop efficient methodologies for data and feature valuation in symbolic-numeric tasks. In essence, SNIP's contributions extend far beyond its current scope, encouraging a future filled with cross-disciplinary innovations.

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

# Appendix

## A  Pre-training Data Details

We provide additional details regarding the pre-training data employed for pre-training SNIP. In our approach, SNIP is pre-trained on a large synthetic dataset of paired numeric and symbolic data, utilizing the data generation technique from [8]. Each example consists of a set of $N$ points $(\boldsymbol{x}, y) \in \mathbb{R}^{D+1}$ and an associated mathematical function $f(\cdot)$, such that $y = f(\boldsymbol{x})$. These examples are generated by first sampling a function $f$, followed by sampling $N$ numeric input points $\boldsymbol{x}_i; i = 1, \ldots, N \in \mathbb{R}^D$ from $f$, and then calculating the target value $y_i = f(\boldsymbol{x}_i)$.

### A.1  Sampling of functions

To generate random functions $f$, we employ the strategy outlined in [8, 3], building random trees with mathematical operators as nodes and variables/constants as leaves. This process includes:

**Input Dimension Selection.** We begin by selecting the input dimension $D$ for the functions from a uniform distribution $\mathcal{U}(1, D_{max})$. This step ensures variability in the number of input variables.

**Binary Operator Quantity Selection.** Next, we determine the quantity of binary operators $b$ by sampling from $\mathcal{U}(D-1, D+b_{max})$ and selecting $b$ operators randomly from the set $\mathcal{U}(+, -, \times)$. This step introduces variability in the complexity of the generated functions.

**Tree Construction.** Using the chosen operators and input variables, we construct binary trees, simulating the mathematical function's structure. The construction process is performed following the method proposed in [8, 3].

**Variable Assignment to Leaf Nodes.** Each leaf node in the binary tree corresponds to a variable, which is sampled from the set of available input variables ($x_d$ for $d = 1, \ldots, D$).

**Unary Operator Insertion.** Additionally, we introduce unary operators by selecting their quantity $u$ from $\mathcal{U}(0, u_{max})$ and randomly inserting them from a predefined set ($\mathcal{O}_u$) of unary operators where $\mathcal{O}_u = [\mathrm{inv}, \mathrm{abs}, \mathrm{pow2}, \mathrm{pow3}, \mathrm{sqrt}, \mathrm{sin}, \mathrm{cos}, \mathrm{tan}, \mathrm{arctan}, \mathrm{log}, \mathrm{exp}]$.

**Affine Transformation.** To further diversify the functions, we apply random affine transformations to each variable ($x_d$) and unary operator ($u$). These transformations involve scaling ($a$) and shifting ($b$) by sampling values from $D_{\text{aff}}$. In other words, we replace $x_d$ with $ax_d + b$ and $u$ with $au + b$, where $(a, b)$ are samples from $D_{\text{aff}}$. This step enhances the variety of functions encountered during pre-training and ensures the model encounters a unique function each time, aiding in mitigating the risk of overfitting as well as memorization.

### A.2  Sampling of datapoints

Once have generated a sample function $f$, we proceed to generate $N$ input points $x_i \in \mathbb{R}^D$ and calculate their corresponding target value $y_i = f(x_i)$. To maintain data quality and relevance, we follow the guidelines from [8], which include: *Discarding and Restarting:* If any input point $x_i$ falls outside the function's defined domain or if the target value $y_i$ exceeds $10^{100}$, we discard the sample function and restart the generation process. This ensures that the model learns meaningful and well-behaved functions. *Avoidance and Resampling:* Avoidance and resampling of out-of-distribution $x_i$ values provide additional insights into $f$ as it allows the model to learn its domain. his practice aids the model in handling input variations. *Diverse Input Distributions:* To expose the model to a broad spectrum of input data distributions, we draw input points from a mixture of distributions, such as uniform or Gaussian. These distributions are centered around $k$ randomly chosen centroids, introducing diversity and challenging the model's adaptability.

The generation of input points involves the following steps:

**Cluster and Weight Selection.** We start by sampling the number of clusters $k$ from a uniform distribution $\mathcal{U}(1, k_{max})$. Additionally, we sample $k$ weights $\{w_j \sim \mathcal{U}(0, 1)\}_{j=1}^k$, which are normalized to $\sum_j w_j = 1$.

**Cluster Parameters.** For each cluster, we sample a centroid $\mu_j \sim \mathcal{N}(0, 1)^D$, a vector of variances $\sigma_j \sim \mathcal{U}(0, 1)^D$, and a distribution shape $D_j$ from $\{\mathcal{N}, \mathcal{U}\}$ (Gaussian or uniform). These parameters define the characteristics of each cluster.

**Input Point Generation.** We sample $[w_j N]$ input points from the distribution $D_j(\mu_j, \sigma_j)$ for each cluster $j$. This sampling with different weights from different distributions ensures the sampling of a diverse set of input points with varying characteristics.

**Normalization.** Finally, all generated input points are concatenated and normalized by subtracting the mean and dividing by the standard deviation along each dimension.

# B    Pre-training Implementation Details

## B.1    Model Design Details

**Numeric Encoder.**    The numeric encoding mechanism of our SNIP closely follows the design presented by [8], as highlighted in Sec. 3. Firstly, for each instance in a given batch, the encoder receives $N = 200$ numeric input points, $(\boldsymbol{x}, \boldsymbol{y})$, from a space $\mathbb{R}^{D+1}$. Each of these points is tokenized into a sequence of length $3(D + 1)$. An embedding module maps these tokens into a dense representation with an embedding size of $d_{\text{emb}} = 512$. The sequences are then processed in the embedder module by a 2-layer feedforward neural network. This network projects input points to the desired dimension, $d_{\text{emb}}$. The output from the embedder is passed to a Transformer encoder, a multi-layer architecture inspired by [2]. Our specific implementation has 8 layers, utilizes 16 attention heads, and retains an embedding dimension of $512$. A defining characteristic of our task is the permutation invariance across the $N$ input points. To accommodate this, we've adopted the technique from [8], omitting positional embeddings within the numeric Transformer encoder. In our design, this specialized encoder variant is termed $Enc^V$. The representation generated at the $l$-th layer of the encoder is represented as $\boldsymbol{V}_l$. The process can be summarized as $\boldsymbol{V}_l = Enc_l^V(\boldsymbol{V}_{l-1})$. Here, the index $l$ spans from 1 to $L_V$, where $L_V = 8$ denotes our encoder's total layers. Post encoding, for each instance in the batch, the numeric encoder's sequence outputs, $\boldsymbol{V}_{L_V} \in \mathbb{R}^{N \times d_{\text{emb}}}$, are compressed into a representation for the whole sequence, $\boldsymbol{Z}_V \in \mathbb{R}^{d_{\text{emb}}}$. This representation captures the essence of the entire numeric sequence and is achieved through an attention-pooling mechanism, detailed in Sec. 3.1.

**Symbolic Encoder.**    Our SNIP's symbolic encoding component draws inspiration from the model used in [3], as highlighted in Sec. 3. This encoder is designed to process mathematical symbolic expressions with a maximum length of 200. These expressions encapsulate the true functional relationships underlying the numeric data fed to the numeric encoder. The expressions are tokenized using a prefix order tree traversal. We employ the vocabulary defined by [8], crafted to comprehensively represent mathematical equations. It includes symbolic entities like variables and operators, along with numeric constants. Constants are tokenized into three parts, consistent with the tokenization method outlined in Sec. 3.1. Sequence boundaries are indicated with special tokens $[\langle BOS \rangle]$ and $[\langle EOS \rangle]$. Tokens are transformed into dense vectors of dimension $d_{\text{emb}} = 512$ using an embedder module. This module essentially functions as an embedding matrix for the employed vocabulary. To maintain uniform input lengths, sequences are padded to a maximum length of $M = 200$ and then projected to the desired embedding dimension. This dimensionality is aligned with the numeric encoder's. The embedded sequences are processed through a Transformer encoder, characterized by its multi-layer architecture as described by [2]. Similarly, our specific configuration for this encoder consists of 8 layers, utilizes 16 attention heads, and retains an embedding dimension of $512$. Contrary to the numeric encoder, the sequence order in symbolic expressions holds significance. Consequently, we are including positional embeddings into this Transformer encoder variant. We denote this encoder as $Enc^S$, and its layer-wise representations are articulated as $\boldsymbol{S}_l = Enc_l^S(\boldsymbol{S}_{l-1})$, iterating from layer 1 to the maximum layer $L_S = 8$. Similar to the numeric encoder's approach, the symbolic encoder condenses its Transformer outputs $\boldsymbol{S}_{L_S} \in \mathbb{R}^{M \times d_{\text{emb}}}$ for each expression into a compact representation, $\boldsymbol{Z}_S \in \mathbb{R}^{d_{\text{emb}}}$. This aggregation leverages the attention-pooling technique detailed in Sec. 3.2.

## B.2    Training Details

Following the extraction of coarse representations from both symbolic and numeric encoders, our focus shifts to harmonizing the embeddings from these encoders. The aim is to closely align embeddings representing corresponding symbolic-numeric pairs, while ensuring a discernible distance between unrelated pairs. As discussed in Sec. 3.3, this alignment process leverages a symmetric cross-entropy loss calculated over similarity scores, with the specific approach being informed by a

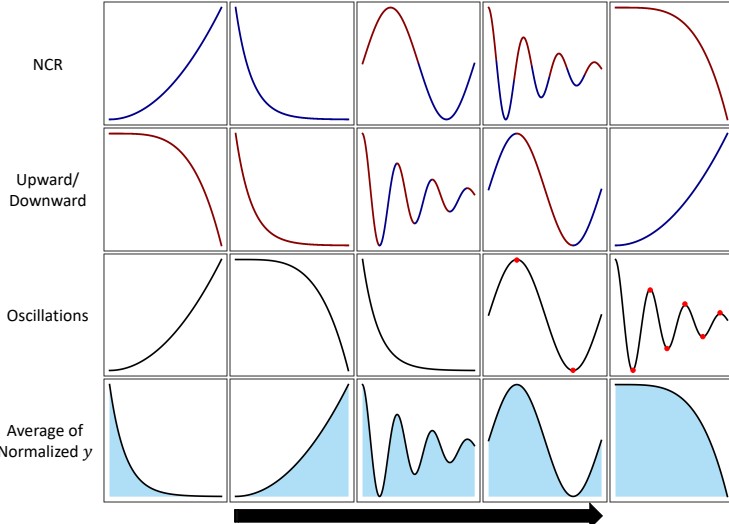

Figure 7: Properties are qualitatively illustrated using five sample functions. Within each row, the plots are arranged according to their respective property values. Colors represent distinct function phases corresponding to the property (e.g., convexity vs. nonconvexity in the first row, upward vs. downward in the second row). Additionally, in the third row, red points highlight instances of change in the y-coordinate.

contrastive loss mechanism. This ensures effective learning of the correspondence between numeric and symbolic data pairs. Our optimization process is facilitated by the `Adam` optimizer, operating on a batch size of $B = 256$ (symbolic, numeric) data pairs. The learning rate initiation is set at a low $10^{-7}$, which is then gradually warmed up to $4 \times 10^{-5}$ over an initial span of 100K steps. Subsequently, in line with the recommendations of [2], we apply an inverse square root decay based on the step count to adjust the learning rate. Our model undergoes training for a total of $\approx 220$ epochs, with each epoch comprising $1,000$ steps. This translates to the processing of $256 \times 1\text{K} = 256\text{K}$ (symbolic, numeric) pair samples for each epoch. Given the on-the-fly data generation mechanism, as highlighted in Sec. A, the cumulative volume of data encountered during pre-training approximates a substantial 60M (symbolic, numeric) pair samples. For training, we utilize 4 GPUs, each equipped with 48GB of memory. Given this configuration, the processing time for a single epoch is approximately two hours.

## C    More details on Pre-trained Latent Space Analysis

### C.1    Properties Definition

In this section, we define the numeric mathematical properties that we use to evaluate the pre-trained SNIP model. The experiments include understanding and predicting numeric properties, i.e., properties that describe the behavior of numeric dataset, from symbolic forms of functions. The formal definitions of these properties are described in the following paragraphs and Fig. 7 qualitatively illustrates what each of the numeric properties represent.

**Non-Convexity Ratio:**    Non-Convexity Ratio (NCR) is defined to quantify the relative convexity (or non-convexity) of the functions as one of the properties depending on the numeric behavior of the functions. Hence, directly predicting this property from the symbolic form of the function is a complex task. To quantify the non-convexity ratio, we employ Jensen's inequality as a fundamental measure [33]. In our approach, we focus on the one-dimensional equations with numeric dataset $\{\boldsymbol{x}, \boldsymbol{y}\}$. Considering a function $f : \mathcal{D} \to \mathbb{R}$ where $\mathcal{D}$ is a convex subset of $\mathcal{R}$, $f$ is a convex function if $\forall x_1, x_2 \in \mathcal{D}$ and $\forall \lambda \in [0, 1]$:

$$f(\lambda x_1 + (1 - \lambda)x_2) \le \lambda f(x_1) + (1 - \lambda)f(x_2).$$

We rely on the training datasets with non-regularly sampled points to calculate the approximate NCR. To this end, we perform multiple trials to examine Jensen's inequality criterion. For each

trial, we randomly select three data points $\{(x_i, f(x_i)), (x_j, f(x_j)), (x_k, f(x_k))\}$ which are sorted based on $x$ in ascending order. The convexity criterion holds on these points if

$$f(x_j) \leq \frac{(x_k - x_j) \cdot f(x_i) + (x_j - x_i) \cdot f(x_k)}{x_k - x_i} + \epsilon, \tag{3}$$

where $\epsilon$ is a very small number ($\epsilon = 10^{-9}$) to avoid numerical precision errors. Therefore, for trial $t$, we define the success as

$$\xi_t = \begin{cases} 1 & \text{if (3) holds,} \\ 0 & \text{otherwise.} \end{cases}$$

Finally, the non-convexity ratio (NCR) is computed over the total number of trials $T$ as

$$NCR = 1 - \frac{1}{T} \sum_{t=1}^{T} \xi_t.$$

Therefore, if a function is always convex over the range of training data points, `NCR=0`, and if it is always non-convex, it would have `NCR=1`. Functions that have both convex and non-convex sections in the range of $x$ will have `NCR` $\in (0, 1)$.

**Upwardness:**  The 'Upward/Downwardness' of a one-dimensional numeric dataset is defined to gauge the proportion of points within the training range where the function exhibits increasing or decreasing behavior. To compute this metric on the sorted dataset $\{\boldsymbol{x_s}, \boldsymbol{f(x_s)}\}$, we examine every consecutive pair of points $\{x_i, x_{i+1}\}$ to determine if they demonstrate an upward or downward trend. We then define $u_i$ as follows:

$$u_i = \begin{cases} 1 & \text{if } f(x_{i+1}) > f(x_i) + \epsilon, \\ -1 & \text{if } f(x_{i+1}) < f(x_i) - \epsilon, \\ 0 & \text{otherwise.} \end{cases}$$

Finally, the upwardness metric `UP` is computed as the average upwardness $\text{UP} = \sum_{i=1}^{N-1} u_i$, where $N$ is the number of points in the dataset. Therefore, if a function is monotonically increasing the range of $x$ in training points, the upwardness measure is $1$, and if it is monotonically decreasing, the metric will be $-1$. Functions that have both sections in the range of $x$ will have $\text{UP} \in (-1, 1)$.

**Oscillation**  For this metric, we aim to quantify the degree of oscillatory behavior exhibited by the numeric data. This is approximated by counting the instances where the direction of $y$ changes. Determining the direction of data points follows a similar process to that of the upwardness metric for each consecutive pair. Thus, we tally the occurrences of direction changes while traversing the sorted dataset. Due to the potential variation in the number of changes, we opt for a logarithmic scale to color the plots.

**Average of Normalized** $y$  The overall behavior of the numeric data points $\{\boldsymbol{x}, \boldsymbol{y}\}$ are better represented when the values of $y$ are scaled to a fixed range (here $(0, 1)$), giving $\{\boldsymbol{x}, \boldsymbol{Y}\}$. The average of the normalized values, $\bar{Y}$ can be a measure to distinguish different numeric behaviors, and it can roughly approximate the numerical integral of the normalized function in the defined range of training $\boldsymbol{x}$.

**Numeric Encoded Representations.**  We show that akin to how symbolic encoded representations are shaped by numeric behaviors, the numeric encoded vectors $\boldsymbol{Z}_V$ are likewise influenced by the symbolic attributes of the corresponding governing equations. To illustrate this, Fig. 3 showcases 2D t-SNE visualizations depicting the learned latent space of SNIP's numeric encoded vectors, color-coded by function (a) complexity and (b) an arbitrarily defined categorization of the functions based on their dominant operators. Further details regarding these two symbolic features are provided below:

*Function Complexity:* Function complexity, as defined in previous works, pertains to the length of the function expressed in prefix order notation,i.e., the number of nodes in the expression tree. Intuitively, functions with a greater number of operators and variables (resulting in longer equations) are considered more complex, often exhibiting correspondingly complex behaviors.

*Function Operator Classes:* Mathematical functions can be broadly classified into different classes based on the operators utilized in their expressions, which in turn influence the behavior of the data they describe. It is important to note that a single function may incorporate multiple operators, contributing to the overall complexity of the data's behavior. Additionally, certain operators within a function may hold more significance than others, exerting greater influence on the range and pattern of the data. To categorize the functions, we employ the following guidelines:

First, we consider a prioritized set of unary operators: $\mathcal{O} = \{\arctan, \tan, \exp, \mathrm{sqrt}, \mathrm{inv}, \cos, \sin, \mathrm{pow3}, \mathrm{pow2}\}$. If a function exclusively employs one of these operators, it is categorized accordingly. For simplicity, we designate both $\mathrm{pow2}$ and $\mathrm{pow3}$ as `Polynomial`, and we employ $\sin$ for both $\sin$ and $\cos$. In the event that a function incorporates more than one operator, it is assigned to the category corresponding to the operator of higher priority. It is worth noting that this categorization may not always perfectly capture the behavior of functions, as an operator with lower priority may potentially exert a more dominant influence than another prioritized operator.

**Annotated Latent Space.**    To have a closer look to the latent space representation, we also analyze several functions with their position in the learned latent space t-SNE visualization. Fig. 8 shows the same t-SNE plot of $\boldsymbol{Z}_S$ (from the symbolic encoder) colored by NCR property and annotated by the numeric behavior (scaled $y$) of some samples. We can observe that the latent space is shaped by both symbolic input $f(\cdot)$ and numeric data, such that closer points have more similar symbolic and numeric features.

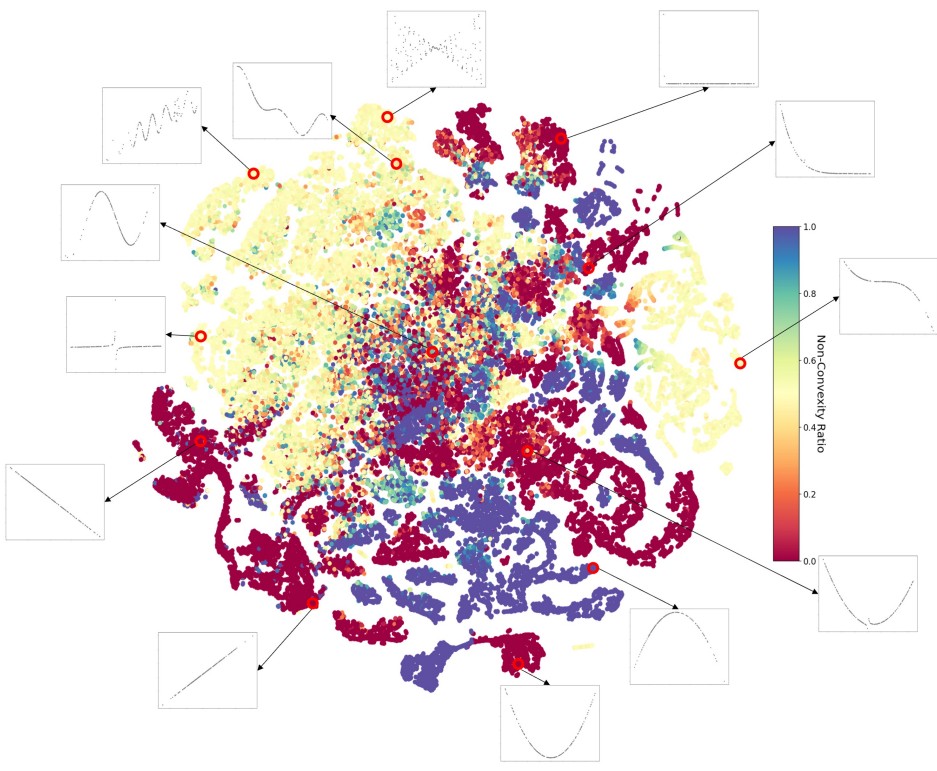

Figure 8: 2D t-SNE plot of the latent space of SNIP symbolic encoded representation $\boldsymbol{Z}_S$ colored by Non-Convexity Ratio property. Several sample equations are plotted with their relative position in the latent space. Both symbolic and numeric aspects of functions affect the latent vectors.

# D    Details of Using SNIP for Cross-Modal Property Prediction

## D.1    Qualitative Results of Property Prediction Models.

Here, we analyze the properties mentioned in the previous section in the latent space. Fig. 9 shows a qualitative comparison of pre-finetuning and post-finetuning latent spaces of SNIP against that of supervised task prediction models, using 2-dimensional t-SNE visualizations of the encoded repre-

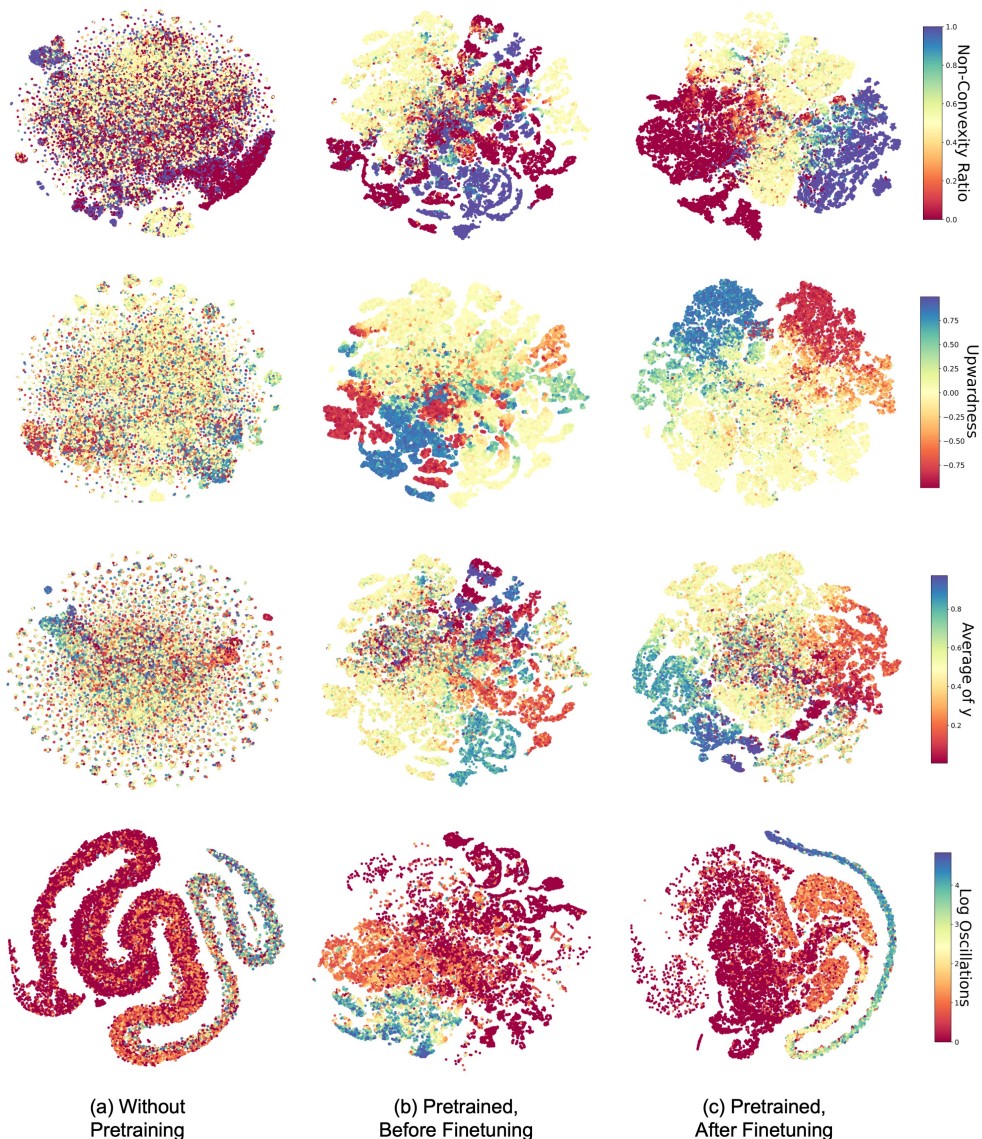

(a) Without Pretraining

(b) Pretrained, Before Finetuning

(c) Pretrained, After Finetuning

Figure 9: 2D t-SNE plots of the encoded representations for the tasks of predicting Non-Convexity Ratio, Function Upwardness, Average of $y$, and Oscillations. The plots compare the **(a)** supervised models without pre-training, **(b)** frozen pre-trained SNIP encoder, and **(c)** fine-tuned SNIP encoders for each task.

sentations. The first two rows (NCR and Upwardness) are replicated from the main body (Fig. 5) for ease of comparison. In each task (row), the plots are colored by the values of the corresponding property. In each task, a training dataset with 10K samples was used to train the model.

The observations from Fig. 9 show that the latent spaces of supervised models (without pre-trained SNIP) are very weakly structured and barely exhibit a recognizable trend for the properties. On the other hand, when the pre-trained SNIP is used, the latent spaces are shaped by the symbolic-numeric similarities of the functions such that numeric properties can be clustered and/or show visible trends in the symbolic encoded representation space $Z_S$. Furthermore, refining the encoder, as shown in Fig. 9(c), leads to more organized latent spaces with distinct linear property trends.

