# OpenReview forum: "SNIP: Bridging Mathematical Symbolic and Numeric Realms with Unified Pre-training"
_NeurIPS.cc/2023/Workshop/AI4Science — NeurIPS2023-AI4Science Poster_

### Official Review · Reviewer_mmHD · 2023-10-24
**Original and potentially impactful work**

**Rating:** 8
**Confidence:** 4

**Review:**

### Summary
The paper introduces SNIP, a multi-modal symbolic-numeric pre-training model designed to understand both symbolic and numeric aspects of mathematical functions. SNIP is trained on a synthetic dataset of paired numeric and symbolic data. It employs t-SNE visualizations to demonstrate that its latent space is influenced by both symbolic and numeric properties. The paper also discusses SNIP's capabilities in property prediction tasks, showing that it outperforms purely supervised models. It uses various training configurations to assess the efficacy of SNIP's learned representations.

#### Pros:
**Originality**: SNIP introduces a new way to understand both symbolic and numeric aspects of mathematical functions, filling a gap in existing research.

**Convincing Probing Tests**: The property probing tests are the highlight for me. They offer a solid, quantitative way to gauge SNIP's capabilities, and I find that really compelling.

**Broad Applications**: The model has potential use-cases in various data-scientific and machine learning tasks.

#### Cons:
**Ignoring Real-world Noise**: The paper doesn't mention how SNIP would handle noise in real-world data, which is a significant limitation. However, this could be simply addressed by adding noise to the numerical data.

**Incomplete end-to-end description**: The paper elaborates on the encoder training but lacks details on the decoding process from latent embedding to symbols or numerics. A good symbolic decoder can make SNIP perform end-to-end regression tasks: observed numerical data -> latent space embedding -> decoded symbols. I would be interested to see how SNIP compares to state-of-the-art symbolic regression methods.

**T-SNE's Limitations**: While the t-SNE visualizations are eye-catching, they feel a bit like window dressing to me. They lack the quantitative rigor that would make them truly convincing.

---

### Official Review · Reviewer_ijis · 2023-10-25
**An interesting paper on mathematical science**

**Rating:** 6
**Confidence:** 4

**Review:**

This paper is very interesting using contrastive learning to solve mathematical problems. The author also conduct experiments to validate their proposed method.

---

### Meta-Review · Area_Chair_py9g · 2023-10-27

**Recommendation:** Accept (Poster)
**Confidence:** 4

**Metareview:**

The study introduces SNIP, a CLIP-like pretraining that bridges symbolic and numeric mathematical domains. The subject matter is captivating. In conclusion, it would be beneficial for the author to include a section on limitations to address feedback from reviewers. Furthermore, providing more specifics on replicating the regression outcomes would be valuable. As those concerns are be addressed by rewriting, I recommend the acceptance of this paper.